# The Predicted Functional Compartmentation of Rice Terpenoid Metabolism by *Trans*-Prenyltransferase Structural Analysis, Expression and Localization

**DOI:** 10.3390/ijms21238927

**Published:** 2020-11-25

**Authors:** Min Kyoung You, Yeo Jin Lee, Ji Su Yu, Sun-Hwa Ha

**Affiliations:** Department of Genetic Engineering and Graduate School of Biotechnology, College of Life Sciences, Kyung Hee University, Yongin 17104, Korea; yeojin@khu.ac.kr (Y.J.L.); dbwltn1025@khu.ac.kr (J.S.Y.)

**Keywords:** functional compartmentation, in silico structural analysis, rice (*Oryza sativa*), subcellular localization, topology-based working model, *trans*-prenyltransferase

## Abstract

Most terpenoids are derived from the basic terpene skeletons of geranyl pyrophosphate (GPP, C_10_), farnesyl-PP (FPP, C_15_) and geranylgeranyl-PP (GGPP, C_20_). The *trans*-prenyltransferases (PTs) mediate the sequential head-to-tail condensation of an isopentenyl-PP (C_5_) with allylic substrates. The in silico structural comparative analyses of rice *trans*-PTs with 136 plant *trans*-PT genes allowed twelve rice PTs to be identified as GGPS_LSU (OsGGPS1), homomeric G(G)PS (OsGPS) and GGPS_SSU-II (OsGRP) in Group I; two solanesyl-PP synthase (OsSPS2 and 3) and two polyprenyl-PP synthases (OsSPS1 and 4) in Group II; and five FPSs (OsFPS1, 2, 3, 4 and 5) in Group III. Additionally, several residues in “three floors” for the chain length and several essential domains for enzymatic activities specifically varied in rice, potentiating evolutionarily rice-specific biochemical functions of twelve *trans*-PTs. Moreover, expression profiling and localization patterns revealed their functional compartmentation in rice. Taken together, we propose the predicted topology-based working model of rice PTs with corresponding terpene metabolites: GPP/GGPPs mainly in plastoglobuli, SPPs in stroma, PPPs in cytosol, mitochondria and chloroplast and FPPs in cytosol. Our findings could be suitably applied to metabolic engineering for producing functional terpene metabolites in rice systems.

## 1. Introduction

More than 800,000 terpene metabolites have been analyzed from diverse living organisms [1]. The largest groups of terpene metabolites have evolved from various photosynthetic higher plants to help allow their survival and adaptation in various ecoregions [2]. The biosynthesis of terpenoids begins with the generation of the building blocks of dimethylallyl pyrophosphate (DMAPP, C_5_) and its isomer isopentenyl-PP (IPP, C_5_) through either the 2-C-methyl-D erythritol-4-phosphate (MEP) pathway in plastids or the mevalonate (MVA) pathway in cytoplasm [3]. The diversity of terpene backbone polymers is firstly determined by *cis*- or *trans*-prenyltransferases (PTs) that mediate consecutive condensation of IPPs with diverse allylic PP substrates, including condensation of DMAPP into *cis*- or *trans*-stereoisomers [4,5]. These compounds are then processed by the terpene synthase (TPS) family proteins into the specified terpene products through modification including cyclization, reduction and branching [6]. On this account, *cis*- or *trans*-prenyltransferases (PTs) are regarded as the first key factors that determine the length of terpene skeletons and limit the diversity of final terpene metabolites. In plants, there have been six kinds of *trans*-PTs reported, including geranyl-PP (GPP, C_10_) synthase (GPS), farnesyl-PP (FPP, C_15_) synthase (FPS), geranylgeranyl-PP (GGPP, C_20_) synthase (GGPS), geranylfarnesyl-PP (GFPP, C_25_) synthase (GFPS), solanesyl-PP (SPP, C_45_) synthase (SPS) and polyprenyl-PP (PPP, C_>50_) synthase (PPS) which mediate sequential head-to-tail condensation of IPP with allylic substrates including DMAPP, GPP, FPP and GGPP, as shown in Appendix A. These *trans*-PTs function as the major metabolic branch points in terpenoid metabolism containing most essential terpenoids such as sterols, gibberellic acids (GAs), chlorophyll, tocopherols, carotenoids, abscisic acid (ABA) and strigolactones (SLs), as well as various functional terpene metabolites for medicinal or industrial applications [7]. On the other hand, the *cis*-PTs mainly take FPP and GGPP among *trans*-isoprenyl-PPs as major allylic donors to sequentially conjugate IPP in a *cis*-orientation and elongate them into long-chain polyprenyl-PP (C_n>50_) [8]. Several *cis*-isoprenyl-PPs, such as neryl-PP (NPP, C_10_), z,z farnesyl-PP (z,z FPP, C_15_) and nerylneryl-PPs (NNPP, C_20_), have been limitedly identified in a few plant systems, such as tomato [9], but the functions of *cis*-PTs are largely unknown still.

Previously, plant *trans*-PTs were classified into five subgroups as FPS (a), SPS (b), PPS (c), GGPS/GFPS/PPS (d) and the small subunit (SSU) of GPS (e) by Jia and Chen [5]) or six clades as GPS/GGPS, SSU-II, SSU-I, GPS/PPS, SPS and FPS by Zhou et al. [10]. Additionally, Wang and Dixon [11]) classified GPS/GGPS families into four clades as homodimeric GPS, SSU-I, SSU-II and large subunit (LSU)/GGPS. However, even in these previous classification systems, the classification of rice *trans*-PTs has not yet been clear, so a comprehensive study on it seems very necessary. Several conserved motifs for plant *trans*-PTs have been characterized in several plant species: IPP-binging motifs as substrate binding sites [12], one or two aspartate-rich motifs (DDX_(2–4)_D) as the first aspartate-rich motif (FARM) and the second aspartate-rich motif (SARM) as being critical to the catalytic activities [4] and CxxxC (x is any hydrophobic amino acid residue) motifs critical for the physical interaction between GGPS_LSU and GGPS_SSU families to alter or enhance GPS/GGPS enzyme activities [10,11]. Furthermore, the “three floors” model has been proposed as a very useful tool to elucidate the *trans*-isoprenyl-PP elongation mechanisms and to predict the characteristic activity of *trans*-PTs in determining the chain length of the linear isoprenoid backbone [13]. This model suggests that the molecular weight of amino acid residues could be a critical factor for determining the chain length with in vitro biochemical validation assays because some bulky residues, such as Y and F, in the floor’s positions of enzyme pockets of *trans*-PTs containing FARM and SARM domains might disturb the elongation. This inhibition would limit the chain length of *trans*-isoprenyl-PPs, whereas some light residues such as G and A might allow the elongation to proceed, producing the longer chain lengths of *trans*-isoprenyl-PPs.

Meanwhile, subcellular localizations have been considered important in predicting the biological roles of *trans*-PTs in a compartment-dependent manner. Most intensively in Arabidopsis, various localizations of eleven GGPSs and four SPSs/PPSs into different organelles such as cytosol, endoplasmic reticulum (ER), peroxisome, mitochondria and chloroplasts have supplied meaningful information to elucidate the in planta function of PTs: Eight chloroplast AtGGPSs (AtGGPS2, 6, 7, 8, 9, 10, 11 and 12) for the biosynthesis of GAs, chlorophyll, tocopherols and carotenoids [14,15], three mitochondrial/ER AtGGPSs (AtGGPS1, 3, 4) for the production of ubiquinone [14], a mitochondrial/chloroplast AtPPS (initially known as AtGPS, Bouvier et al. [16]) for the generation of ubiquinone and plastoquinone [17] and chloroplast AtSPS1 and 2 for the biosynthesis of plastoquinone [18]. These studies suggest that the functional topology of plant PTs might be a key determining factor for predicting the dynamic network of terpenoid metabolism, together with their timely and spatial expression patterns [19].

Among rice *trans*-PTs, some biochemical functions of seven PTs have been elucidated: OsGPS, OsGGPS1 and rice GGPS recruiting protein (OsGRP) [10], OsSPS1 and 2 [20], OsSPS3 [21] and OsFPS1 [22]. However, biochemical functions of the remaining rice *trans*-PTs remain undefined. Moreover, no topology-based study has been conducted to offer insight into rice *trans*-PTs overall. Hence, we identified all of the rice *trans*-PT genes from the rice whole genome database, constructed a new phylogenetic tree showing better classification with extended plant *trans*-PT sequence information and predicted biochemical functions by matching sequences with known conserved motifs and the “three floors” model. Taken together with expression patterns and subcellular localizations, we predicted a topology-based working model of rice *trans*-PTs as a unique system for terpenoid metabolism in rice plants.

## 2. Results

### 2.1. Phylogenetic Tree Analysis of Rice Trans-Prenyltransferases

Twelve genes of rice *trans*-PTs were identified from the rice whole genome database, through the BLAST (Basic Local Alignment Search Tool) search engine. To more precisely predict their function, we performed further informatics searches of molecular databases and constructed an extended phylogenetic tree with 136 plant PTs collected from 13 monocot plant species, including rice, corn, wheat, barley, sorghum and 39 dicot plants including major crops (Figure 1, Appendix A). Three groups of Group I (GPS/GGPS/GFPS family), II (SPS/PPS family) and III (FPS family) were classified to be further assorted into six (A to F), two (A and B) and three (A to C) subgroups, respectively (Figure 1). Among them, twelve of the rice PTs were classified, with three in Group I, four in Group II and five in Group III and their gene information was summarized with some corrections (Table 1).

First in Group I, OsGGPS1 was classified as GGPS_LSU (subgroup A) along with several uncharacterized monocot and dicot PTs, separately from most known GGPS homologs of hop (Hl ACQ90682), *Arabidopsis* (AtGGPS11, AT4G36810) and peppermint (Mp AAF08793). These known GGPS homologs were classified as GGPS_LSU (subgroup A), which has functions as homomeric GGPS or heteromeric GPS when formed in a complex with GGPS_SSUs [11,23]. Additionally, five *Arabidopsis*-specific GGPSs were classified as GFPS (subgroup B) between two LSU subgroups, adding further support for the previous suggestion that they evolved from GGPS_LSU to biosynthesize the longer product of GFPP (C_25_) [13].

**Table 1 ijms-21-08927-t001:** The summarized molecular information of rice *trans*-prenyltransferases (PTs).

Groups	Names	Gene IDs	Protein IDs	Size (aa)	WoLF PSORT ^a^	Enzyme Activity	Ref.
I_GPS/GGPS-C	OsGPS	Os01g14630	BAS71331	416	Chl /Mito	GPP/GGPP synthase	[10]
I_GPS/GGPS-A	OsGGPS1	Os07g39270	BAT02334	366	Chl /Mito	GGPP synthase	[10]
I_GPS/GGPS-F	OsGRP	Os02g44780	XP_015626863	342	Chl	GGPS-recruiting protein	[10]
II_SPS/PPS-B	OsSPS1	Os06g46450	BAS99121	430	Mito/Chl	SPP/PPP synthase	[20]
II_SPS/PPS-A	OsSPS2	Os05g50550	Q75HZ9	403	Chl/Mito	SPP synthase	[20]
II_SPS/PPS-A	OsSPS3	Os12g17320	XP_015620200	414	Chl/Mito	SPP synthase	[21]
II_SPS/PPS-B	OsSPS4	Os08g09370	XP_015648843	399	Chl/Mito	The predicted-PPP synthase	-
III_FPS-B	OsFPS1	Os01g50760	BAS73911	353	Cyto	The predicted-FPP synthase	[22]
III_FPS-B	OsFPS2	Os05g46580	BAS95151	355	Cyto	The predicted-FPP synthase	[24]
III_FPS-B	OsFPS3	Os01g50050	XP_015621770	356	Cyto/Nuc	The predicted-FPP synthase	-
III_FPS-A	OsFPS4	Os04g56230	MT793647	354	Cyto	The predicted-FPP synthase	-
III_FPS-A	OsFPS5	Os04g56210	BAS91413	392	Cyto/Nuc	The predicted-FPP synthase	-

^a^, WoLF PSORT: https://wolfpsort.hgc.jp/; Chl, chloroplast; Cyto, cytosol; Mito, mitochondria; Nuc, nucleous.

Secondly, OsGPS was classified as G(G)PS (subgroup C) with two monocot PTs from corn (*Zea may*, ABQ85648) and sorghum (*Sorghum bicolor*, XP_021311468), but separately from the well-characterized homomeric GPS (subgroup D) originating from Grand fir (*Abies grandis*, AAN01134, Burke and Croteau [25]) and Norway spruce (*Picea abies*, ACA21458, Schmidt and Gershenzon [26]). A G(G)PS (subgroup C) included four AtGGPSs (AtGGPS1, 3, 4 and 8) that have GGPS/PPS activities mainly when using FPPs as allylic substrates [13], supposing that monocot PTs of subgroup C containing OsGPS might be a type of PT family commonly using FPPs as allylic substrates. The third rice PT, OsGRP, was classified with other uncharacterized monocot PTs in GGPS_SSU-II (subgroup F) containing AtGGPS12 (At4G38460, Chen et al. [27]), but separately from GGPS_SSU-I (subgroup E) including the heterodimeric GPS of hop (Hl ACQ90681) and the heterotetrameric GPS of peppermint (Mp AAF08792) that form a complex with hop (Hl ACQ90682) or peppermint (Mp AAF08793) in GGPS_LSU subgroup as counterparts [11,23]. Collectively in Group I, the GPS/GGPS/GFPS family have evolved to have six subgroups of A–F in the plant kingdom, resulting in offering different preferences of enzymatic activities for the production of isoprenyl-PPs. Rice has evolved to have a limited number of GPS/GGPS/GFPS genes as there are only three, in contrast to the existence of twelve GPS/GGPS/GFPS genes in *Arabidopsis*.

Next for Group II, as the SPS/PPS family, four rice PTs were identified in the whole genome, including the already known OsSPS1 and 2 [20] and OsSPS3 [21] and the new OsSPS4 in this study (Figure 1 and Table 1). The phylogenetic tree showed that OsSPS2 and 3 in a subgroup A were more closely related to AtSPS1 and 2 (ABF58968 and ABI54337, Jun et al. [18]) and OsSPS1 and 4 in a subgroup B of AtPPS (AAW39025, Hsieh et al. [17]), suggesting that the families of subgroup A and B are expected to play differential roles as long-chain *trans*-PTs. Moreover, both subgroup A and B are further categorized into monocot and dicot plants and the independent evolution in the plant kingdom is indicated.

In Group III, as the FPS family, five rice PTs were identified and designated according to sequence similarity, with *Arabidopsis* FPS1 (AAL34286, Closa et al. [28]), a known OsFPS1 (70.5%, Sanmiya et al. [22]) and four new rice FPS genes in this study: OsFPS2 (66.5%), OsFPS3 (62.4%), OsFPS4 (52.5%), OsFPS5 (50.1%) (Figure 1 and Table 1). Among three subgroups of A, B and C in Group III, none of the dicot FPS were classified in the subgroups of A or B, and none of monocot FPS were observed in subgroup A, showing evolutionary dissociation between monocot and dicot plants for the adaptation to environmental conditions. Rice FPSs belonged to two subgroups: OsFPS4 and 5 in subgroup A and OsFPS1 to 3 in subgroup B. Interestingly, wheat (*Triticum aestivum*) FPSs [29] are classified into both subgroups, like rice, but all of maize FPSs [30] belong to a subgroup B, unlike rice. Taken together, rice has evolved to have a high copy number of FPS family genes, as there are five, compared to two copies reported in *Arabidopsis*. Subgroup A genes including OsFPS4 and 5 may have evolved to have some unique roles in several monocot plants including rice, barley (*Hordeum vulgare*), sorghum (*Sorghum bicolor*) and wheat, compared to other plant FPS subgroups.

### 2.2. Comparative Sequence Alignments of Rice Trans-Prenyltransferases in the Highly Conservative Regions

To support the phylogenetically predicted functions by matching them with known conserved motifs, multiple alignment of rice *trans*-PTs along with representatives of other *trans*-PTs in each subgroup showed relatively good conservation. However, there were several exceptions in four regions (I-IV), including core sites in regions I (GKR) and II (H/QX_(15–17)_RR) for IPP-binding, aspartate-rich motifs in regions II (FARM, DDX_(2–4)_D) and IV (SARM, DDX_2_D) for catalytic activity, which have been reported as three highly conserved regions for critical function as common PTs, and two CxxxC motifs in regions I and IV for forming the heteromeric complex of GGPS_LSU and _SSU (Figure 2A, Appendix A). The first IPP-binding site showed major variation only in Group I PTs (Figure 2A): Monocot families containing OsGRP (GGPS_SSU-II, subgroup F) had the insertion of seven amino acids between the first G and the third K residues, but dicot families of GGPS_SSUs contained one X insertion between G and K residues (GGPS_SSU-II) or no GKR motif (GGPS_SSU-I), suggesting that these peculiar properties might be largely responsible for their differential roles of GGPS_SSUs. Group II PTs displayed conservation (GKR/Q) between subgroup A and B as no insertion and all of Group III PTs contained the same motif (GK.R), as shown in Figure 2B,C. The second IPP-binding site was well conserved within Group I (HX_17_RR), Group II (HX_15_RR) and Group III (QX_15_RR). These results suggest that the first IPP-binding site might have crucial roles for the differential functions among plant PTs containing GGPS_SSU families. In cases of two aspartate-rich motifs, there are also major variations only in Group I: A Mp AAF08792 in subgroup E was defective in both FARM and SARM, and all of PTs belonged to subgroup F, including an OsGRP, have no fully conserved SARM due to absence of the last D residue, supporting lack of catalytic activity of GGPS_SSUs alone as the second representative characteristic. Lastly, two CxxxC motifs were presented only in Group I, but not in Groups II and III (Figure 2).

In particular, the first C residue of a CxxxC motif in a region I was varied in three monocot PTs, including an OsGPS in subgroup C as an AxxxC motif, supposing a different type of physical interaction, but the second CxxxC motif in a region IV conservatively existed expectedly only in subgroups E and F. Collectively, two rice PTs in Group I showed remarkably different patterns: The first IPP binding motif (GKR) of OsGRP had additional amino acid residues between the G and K residues and OsGPS contained an AxxxC motif in region I, instead of a CxxxC motif (Figure 2A). In contrast, the other rice PTs had no modified residues in those highly conserved domains to show the general patterns of plant PT families.

To infer the biochemical function of rice *trans*-PTs on using the “three floors” model suggested by Wang et al. [13], the determinant amino acid residues corresponding to Floor 1, Floor 2 and Floor 3 in the elongation channels were individually displayed as green, red and yellow (Figure 2 and Appendix A). To derive the specified patterns using the three floor motifs among plant *trans*-PTs, we constructed color-based HeatMap graphs according to the ranges of the molecular weights of amino acid residues (Figure 3).

Interestingly, the color patterns of all floors were collectively distinguishable among groups and subgroups. The better matches with the phylogenetic patterns of Groups I–III were displayed in Floor 1 and 3 (Figure 3). Group I was full of residues in all three Floors, but the third residue of Floor 1 was defective in Group III and the second residue of Floor 3 was filled with no amino acid in Group II, a heavy amino acid of F or Y (Molecular weight (MW): >165.2) in Group I and lower weight residues such as I or V (MW: < 131.2) in Group III, supposing the differential biochemical roles among Groups I–III. In Group I, diverse but distinguishable variations in residues of Floor 1, 2 and 3 were displayed among the six subgroups, matching with the phylogenetic patterns (Figure 1 and Figure 3A).

The better matches with the classification patterns of subgroups were observed in Floor 1 and 2 of Group I and Group II (Figure 3). GGPS_LSUs (subgroup A) have the same amino acid residue weight in each position of Floor 1 and Floor 3, but GGPS_LSU (subgroup A), which includes OsGGPS1, has a distinguishing light V residue (MW: 117.1) at the first position of Floor 2, instead of a heavy I residue (MW: 131.2) and GFPS (subgroup B) has a light S residue at the first position in Floor 1, supposing their differentially biochemical functions. Moreover, the two subgroups of GPS (D) and G(G)PS (C) showed the highest variation in Floor 1 and 2, compared with other PTs, but interestingly, two gymnosperm homomeric GPSs (Ag AAN01133 and Pa ACA21458) and one rice GPS (OsGPS) had the same residues in Floor 1, supposing their similar biochemical functions. Lastly, two GGPS_SSUs (subgroup E and F) have the same heavy M residue at the third position of Floor 1 and GGPS_SSU-II (subgroup F), which includes OsGRP, has specifically a heavy F residue at second position in Floor 2. In Group II, subgroup A, which includes OsSPS2 and 3, contained an F residue (MW: 165.2) in each of Floor 1 and 2, while subgroup B, which includes OsSPS1 and 4, had an L residue (MW: 131.2) in the same position (Figure 3B). In Group III, the first two residues of Floor 1 in subgroup A, which includes OsFPS4 and 5, were observed to be quite different as the light residues S and A (MW: <105.1), compared with the heavy residues of F and Y in subgroup B and C (Figure 3C).

### 2.3. Spatial and Developmental Expression Patterns of Rice Trans-Prenyltransferases Genes

To profile the expression patterns of rice *trans*-PT genes, transcript levels were examined by qRT-PCR using rice RNAs from the leaves, roots and florets at stages of seedling, early and late vegetative, reproductive and mature seeds (Figure 4). The *OsGPS* gene showed relatively low expression, with the highest peak in leaves at reproductive stage. The expression patterns of *OsGGPS1* and *OsGRP* genes were very similar, displaying the highest levels in leaves at the early vegetative stage. This expression pattern revealed leaf-preferential expression of three rice PTs of Group I (Figure 4A), reflecting the property to form a heteromeric complex between *OsGGPS1* and *OsGRP* [10]. Among four rice PTs of Group II, *OsSPS1* and *OsSPS4* in a subgroup B were expressed at significantly low levels in all tissues. The highest peak of *OsSPS4* in roots was at the early vegetative stage and *OsSPS2* and *OsSPS3* in subgroup A were expressed mainly in leaves of all developmental stages, with the highest peaks at the late vegetative stage (Figure 4B), showing similar expression patterns within the same subgroup. Moreover, the highest expression peak of *OsSPS2* in leaves proposes that an OsSPS2 might play a major role as an SPS in the leaves of rice plants. Among five rice PTs of Group III, *OsFPS1* and *3* in subgroup B were expressed with preference to leaves at all developmental stages, while *OsFPS2*, another gene in subgroup B, was mainly expressed in roots of all developmental stages. They were all slightly expressed in florets and seeds. Unlike these, *OsFPS4* of subgroup A was exclusively expressed in leaves of all developmental stages (Figure 4C). Lastly, the expression of *OsFPS5* was also analyzed, but was not detected in our data set at all and further investigated in a RiceXPro database to find out that it is quite limitedly expressed in the early stage of embryo (Appendix A).

### 2.4. Expression Patterns of Rice Trans-Prenyltransferases According to the Treatment of Major Hormones, meJA, GA and ABA

Since the production of terpene metabolites have been reported to be closely related to responses against various plant hormones, such as methyl jasmonate (meJA) [31,32], gibberellic acid (GA) [33] and abscisic acid (ABA) [34], the transcriptional response of rice *trans*-PTs to meJA, GA and ABA treatments was investigated. The hormone-treated rice whole plants were examined by qRT-PCR with three inducible marker genes; rice terpene synthase 19 (*OsTPS19*, Chen et al. [35]) for meJA, rice gibberellin 2-oxidase 3 (*OsGA2ox3*, Chu et al. [36]) for GA and rice phytoene synthase 1 (*OsPSY1*, Welsch et al. [37]) for ABA (Figure 5). After meJA treatment, the expression of *OsGPS*, *OsGGPS1*, *OsSPS1* and *OsFPS1* and *3* were remarkably induced and that of *OsGRP*, *OsSPS2* to *4,* and *OsFPS2* and *4* were suppressed. After GA treatment, the expression of *OsGGPS1*, *OsGRP*, *OsSPS1*, *OsFPS1* and *OsFPS4* were induced within 30 min and that of *OsGPS*, *OsSPS2* to *4* and *OsFPS3* were increasingly suppressed over time. Post ABA treatment, the expression of *OsGGPS1*, *OsSPS2* and *OsFPS3* and *4* were induced beginning at 2 h and the transcript levels of *OsGPS*, *OsGRP* and *OsFPS2* were quite reduced. Collectively, expression patterns of rice PTs showed differential responses to different hormones and the expressions of *OsGPS* and *OsGGPS1* were strongly induced with *OsTPS19* monoterpene synthase after meJA treatment, supposing that they might play an important role for monoterpene production.

### 2.5. Subcellular Compartmentation of Rice PTs in Rice Protoplasts

To investigate the subcellular localization of rice PTs, synthetic green fluorescent protein (sGFP) was fused to their C-termini and transfected into the leaf protoplasts prepared from 10-day-old rice seedlings (Figure 6). For rice Group I families, green fluorescence for OsGPS:sGFP, OsGGPS1:sGFP and OsGRP:sGFP were similarly observed to be spread in stroma with strong spot-like patterns inside chloroplasts (Figure 6A). Two latter ones were also strongly detected in stroma (Appendix A). To verify the subcellular structure of these spot-like patterns, an OsPSY2:mCherry construct, which is targeted into plastoglobules [38], was individually co-transfected. Green fluorescence exactly merged with red fluorescence, indicating that OsGPS, OsGGPS1 and OsGRP proteins localize mainly to both plastoglobules and stroma of chloroplasts. For rice Group II PTs, OsSPS1:sGFP, OsSPS2:sGFP, OsSPS3:sGFP and OsSPS4:sGFP were transfected into rice protoplasts and their green fluorescence resulted in different patterns in localization (Figure 6B). OsSPS1 was localized in two places of stroma and strong spot-like patterns outside chloroplasts. To verify the cellular compartmentation of OsSPS1, the Red CMXRos fluorescent signals of MitoTracker as a mitochondrial marker probe were presented as yellow signals. These signals matched with green fluorescence of OsSPS1:sGFP, indicating the dual localization to stroma and mitochondria. OsSPS2:sGFP and OsSPS3:sGFP were localized to stroma of chloroplasts. The green fluorescence for OsSPS4:sGFP was observed as big spot-like patterns, but were still in cytoplasm. To verify these unusual signals, three organelle-specific markers of MitoTracker, G-rb for Golgi apparatus and px-rb for peroxisomes with red fluorescence were co-transfected, but did not merge with the green fluorescent signal of OsSPS4:sGFP at all, proposing its cytosolic localization into some kinds of vesicles. For rice Group III PTs, five sGFP fusion constructs with OsFPS1–5 were individually transfected into rice protoplasts (Figure 6C). OsFPS1–4 nicely showed cytosolic localization patterns, but OsFPS5 was aggregated as a few big-spots, but still in cytoplasm (Figure 6C).

## 3. Discussion

Terpene metabolite diversification has allowed plant adaptation to environmental conditions. Accordingly, plant PTs for supplying linear *trans*-isoprenyl-PP backbones have been evolutionarily fine-tuned to result in structural and functional diversity in plant species-specific manners with complexity [40]. To understand the function of rice *trans*-PTs, we comprehensively integrated an extended phylogenetic tree (Figure 1 and Appendix A), core variation of amino acid residues in conserved motifs and a color-based HeatMap display applied to the “three floors” model (Figure 2 and Figure 3) and expression patterns (Figure 4 and Figure 5) and subcellular localizations (Figure 6).

In our phylogenetic tree, GGPS/GGPS/GFPS family (Group I) was subdivided into six subgroups (Figure 1). Rice has relatively simpler system in Group I, with only three PTs, compared to *Arabidopsis* with twelve PTs. Previously in rice, OsGGPS1 (LSU) was reported to have multiple functions as GGPS (67%), GPS (27.2%) and FPS (5.5%) as a homomeric complex and OsGRP (SSU-II) strongly enhanced GGPS function as heteromeric complex with OsGGPS1 [10]. In previous reports, the heteromeric LSU-SSU complexes mediated the enhancement of GPP-biosynthetic activity to produce both GPP and GGPP using DMAPP, GPP and FPP in *Arabidopsis* [15] and hops (*Humulus lupulus*; Wang and Dixon [11]) or enabled GGPS_LSU to biosynthesize only GPP without any GGPP-production, even if the homomeric LSU complexes had GGPP-biosynthetic function, as in snapdragon (*Antirrhinum majus*, Tholl et al. [41]) or no enzymatic function, as in peppermint (*Mentha x piperita*, Chang et al. [23]) and in those plant system, any homomeric GPS has been not reported yet. Considered together, we supposed that the heteromeric LSU-SSU complexes might have evolved to have varied enzymatic preferences for the biosynthesis of GPP and GGPP, depending on the environmental conditions of corresponding plant species and in rice with homomeric GPS, the heteromeric OsGGPS1-OsGRP complex has evolved to enhance GGPP-biosynthetic activity, instead of acquiring GPP-biosynthetic activity.

Previously, OsGPS was reported to have the dual functions of both a GPS using DMAPP and a GGPS using FPP [10] (Table 1). However, our phylogenic tree showed that OsGPS was categorized into a G(G)PS subgroup (C) that shows a close relationship with AtGGPS1, 3, 4 and 8, rather than into a representative homomeric GPS subgroup (D) composed of several well-characterized GPSs from the gymnosperm trees Grand fir and Norway spruce (Figure 1). AtGGPS1 and AtGGPS4 have GGPS activities using FPP exclusively in mitochondria and in ER, respectively, and AtGGPS3 using GPP and FPP and AtGGPS8 using all allylic substrates of DMAPP, GPP, FPP and GGPP contain the PPS activities in ER and chloroplasts, respectively [13]. Considered together with the functions of OsGPS, the subgroup (C) families are suggested to have the common feature of using FPP as an allylic substrate, rather than their final product lengths. As expected, the structural analysis following the “three floors” model showed that all residues of “three floors” were quite varied in a subgroup (C) (Figure 3A), but interestingly, OsGPS had the same residues of Floor 1 as in homomeric GPSs in subgroup D, even if its Floor 2 residues did not match at all. These results support that OsGPS has both structural properties of subgroup C and D, as a new type of a homomeric GPS family. Actually, the expression levels of *OsGPS* were significantly lower than those of *OsGGPS1* or *OsGRP* in all tissues of all developmental stages (Figure 4), but were largely induced by the meJA treatment, with patterns similar to those of *OsTPS19* followed by *OsGGPS1* (Figure 5). Taken together, OsGPS is suggested to have functions on monoterpene/diterpene metabolism as a homomeric GPS using DMAPP or a GGPS using FPP under the condition of JA-related responses.

Localizations of twelve rice PTs were confirmed, with five mainly in chloroplasts, one dually in chloroplasts and mitochondria and six in cytosol in rice cells (Figure 5). Within chloroplasts, OsGPS, OsGGPS1 and OsGRP were mainly localized into plastoglobules, even though they were also observed in stroma (Figure 6A) and OsSPS1 to 3 existed in stroma (Figure 6B). In contrast to other plastidial GPS/GGPS homologous from *Arabidopsis* and *Catharanthus roseus* that are mainly active in stroma [14,42], rice GPS/GGPS families peculiarly seem to take plastoglobules as the main venue for the biosynthesis of GPP and GGPP, and supply the primary substrates of monoterpenes, diterpenes, tetraterpenes and long chain-PPs. As with other plastidial PTs in rice, OsSPS2 and OsSPS3 were mainly observed in stroma of rice protoplasts, but not in plastoglobules (Figure 6B) and the plastidial localization of OsSPS2 was also observed in tobacco leaf cells [20]. Other studies reported that OsSPS1 was targeted only in mitochondria of onion and tobacco leaf cells [20], however, OsSPS1 was dual-localized in both mitochondria and chloroplasts in this study and AtPPS1 (AAW39025) with close relation to OsSPS1 in our phylogenetic tree was similarly dual-localized into mitochondria and stroma [43], supporting their similar functions in mitochondria and chloroplasts. Another SPS, OsSPS4, was localized in cytosol with spot-like patterns, regarded as cytosolic vesicles (Figure 6B). Taken together, GPP/GGPPs are supposed to be produced in plastoglobules as main venues and also in stroma and SPP/PPPs are regarded to exist in chloroplasts, mitochondria and cytosolic vesicles.

As summarized in Table 1, GPP/GGPP can be produced by both OsGPS and OsGGPS1 in rice. OsGPS not only biosynthesize mainly GPP, but can also produce GGPP under the limited conditions in which FPP is present as a substrate, and likewise, OsGGPS1 can also produce the limited amount of GPP in addition to GGPP [10]. OsSPS1 and OsSPS2 were reported to produce SPP/PPP by using the primary substrates such as GPP, FPP and GGPP [20], OsSPS3 [21] and OsSPS4 were identified as the putative SPS family proteins. The FPP was predictably produced by five kinds of OsFPSs in this study and two previous studies [22,24]. As the next step, we have further considered the important roles of rice PTs in supplying the primary substrates (GPP/FPP/GGPP) to terpene biosynthetic enzymes. As shown in Figure 5, the expression of OsTPS19 is strongly increased by meJA-treatment with OsGPS, which utilizes GPP as a substrate for producing monoterpenes such as limonene and linalool [35], but it was reported that OsTPS19 existed in stroma [35], in contrast to the localization of OsGPS to PGs (Figure 6). If so, how can GPPs be supplied to OsTPS19 for monoterpene production? Firstly, an OsGGPS1 existing in stroma could be regarded as the GPP-supplier of OsTPS19, which of expression is induced by meJA treatment, and as another possibility, it could be also considered that an OsGPS or OsTPS19 might be translocated into stroma or PGs or GPPs itself might be transported into stroma from PGs. The ABA-induced OsGGPS1 and OsPSY2 are similarly observed in both stroma and PGs (Figure 5 and Figure 6A), suggesting that OsGGPS1 has functions on supplying GGPP to OsPSY1,2,3, considered with the previous report that OsPSY1,2,3 were mainly localized in chloroplast stroma and PGs [44]. Additionally, in a previous study, OsSPS1 preferentially utilized FPP as a main substrate for SPP biosynthesis, compared to GPP or GGPP as minor allylic substrates [20]. As well, OsGPS was reported to limitedly produce GGPP in vitro under the condition that FPP was supplied as an allylic substrate [10]. As the cases stand, the possibility of FPP in chloroplast could not be ruled out, even if the existence of FPP in chloroplasts has been still largely unknown. If so, a question arises of how FPP is supplied to OsSPS1 and OsGPS for SPP and GGPP production in chloroplasts, respectively.

As several possibilities, GPP/FPP could be supplied by GPS activity (27.2%) and FPS activity (5.5%) of OsGGPS1 within chloroplasts or the flux of FPPs from cytosol could be utilized by OsSPS1 and OsGPS, as shown in Figure 7. Moreover, the biosynthetic activity of a plastidial OsSPS2 requires the existence of GPP as an allylic substrate [20], but OsGPS producing GPP in chloroplasts is expressed at a very low level in all rice tissue under the normal conditions (Figure 4). Considered with the expression levels of OsSPS2, the absolute substrate shortage is expected, and OsGGPS1 is possibly supposed to supply GPP to OsSPS2 through its GPS activity (27.2%) for the biosynthesis of SPPs, essential to produce plastoquinone in chloroplasts (Figure 7).

In summary, all rice *trans*-PT genes through a phylogenetic tree using 136 plant *trans*-PT genes were collected, and their biochemical functions were predicted by matching sequences with known conserved motifs and the “three floors” model, taken together with their expression patterns and subcellular localizations. Collectively, we propose a predicted topology-based working model of rice *trans*-PTs to biosynthesize the linear *trans*-isoprenyl-PP backbones in Figure 7, by the following points: Where rice PTs are distributed in rice cells, what kinds of allylic substrates they utilize and what kinds of terpene metabolites are produced in rice. As the major cellular-compartmentations of *trans*-isoprenyl PPs, GPPs for mono-terpenoids are produced by OsGPS in plastoglobules or OsGGPS1 in stroma, GGPPs for di-/tetra-terpenoids are produced by a heteromeric complex between OsGGPS1 and OsGRP or a homomeric OsGGPS1 in both plastoglobules and stroma, SPP/PPP are by OsSPS2 in stroma for plastoquinone, by OsSPS1 in mitochondria for ubiquinone/polyprenols and by OsSPS4 in cytosolic vesicles for dolichols/polyprenols and FPPs are by OsFPS1 in cytoplasm for sesqui-/tri-terpenoids (Figure 7).

In rice, the diverse terpene metabolites have been reported, which include phytohormones, chlorophyll, tocopherols, carotenoids, plastoquinone, ubiquinone, phytosterols and dolichols to play diverse essential roles for plant development and for adaptation to environmental conditions, antimicrobial monoterpenes such as limonene, linalool and terpinene, and other anti-fungal and insect-attractant terpenes such as momilactones, phytoallexins (oryzalexins), bisabolene, caryophyllene, and nerolidol (bactericides) [45,46,47,48]. Our networking model of all predictable rice PTs based on subcellular topology expands our understanding about rice terpenoid metabolism, and if combined with the further identification of enzymatic activity in the near future, it could be very useful to apply to functional terpenoid metabolic engineering in monocot plant systems, including rice.

## 4. Materials and Methods

### 4.1. Amino Acid Sequence Analysis

The deduced amino acid sequences of 136 plant *trans*-PTs genes were collected by BLAST search (https://blast.ncbi.nlm.nih.gov/Blast.cgi) from 13 monocot plant species including corn, wheat, barley, sorghum, 30 dicot plant species including major crops and 9 gymnosperm plant species. A phylogenetic tree of plant *trans*-PTs was generated as a circular tree using ClustalW algorithm and UPGMA method in MEGA 7.0.26 software [49]. The deduced amino acids of the representative PTs in each subgroup of Groups I–III were aligned using the standard parameters of the ClustalW algorithm of MegAlign (DNAStar, Madison, WI, USA) and presented using GeneDoc (DNAStar).

### 4.2. Gene Expression Analysis

Mature seeds of Korean rice “Ilmi” (*Oryza sativa*) were sterilized with 70% ethanol and 2% sodium hypochlorite and washed five times with distilled water. The sterilized seeds were grown on Murashige and Skoog (MS) agar medium under the conditions of 16 h light/8 h dark at 28 °C over 10 days and then the leaves and roots of seedling stage (SS) plants were harvested. For plant preparation of the other two stages, vegetative stage (VS) and reproductive stage (RS), 5 days old healthy seedling plants were transplanted to soil and grown in a greenhouse under conditions of 16 h light/8 h dark at 28 °C [50].

Korean rice (*Oryza sativa* L. cv. Ilmi) seeds were obtained from “National Institute of Agricultural Sciences (South Korea)”. For various hormone treatments, the sterilized seeds were grown on MS agar medium for 3 days and healthy seedling plants were transplanted and grown on four layers of gauze (5 × 5 cm) in the Incu tissue culture frames (model 310071; SPL Lifesciences, Seoul, Korea) containing 45 mL of liquid MS medium (Duchefa, Haarlem, Netherlands) under 16 h light/8 h dark at 28 °C for 7 days, which were prepared by filling the culture frames containing four layers of gauze (5 × 5 cm) with the liquid MS medium (Duchefa) up to the 45 mL-marked line and sterilizing them by autoclave. Plant hormones methyl jasmonic acid (MeJA), gibberellic acid (GA) and abscisic acid (ABA) were prepared as 10 × stock solution of 1 mM and after checking whether 45 mL-liquid MS media were filled in the culture frames, 5 mL of 1 mM hormone solution (10×) were added to each 45 mL of liquid MS medium with gentle stirring. For the 10-day old seedling plants that had been grown on gauze, whole plants containing leaves and roots were sampled at 0, 30 min, 1, 2, 4 and 6 h post treatment into liquid nitrogen for the expression pattern analysis of rice PTs genes.

Total RNA was purified from the frozen powder (100 mg) of samples using an RNeasy Plant Mini Kit (Qiagen, Hilden, Germany) and DNase I (Qiagen) for removal of remnant genomic DNA contamination. The 1st cDNA strand was synthesized using AccuPower^®^ RT Premix (Bioneer, Daejeon, Korea) and cDNA was mixed with SYBR Green Real-time PCR master mix (Bio-Rad, Richmond, CA, USA) according to the manufacturer’s instruction. The quantitative real-time PCR (qRT-PCR) assays were performed with a CFX Connect^™^ Real-Time System (Bio-Rad) under the following conditions: 1 cycle of 3 min at 95 °C, 40 cycles of 15 s at 95 °C and 1 cycle of 30 s at 60 °C [50]. All primers for qRT-PCR were designed as intron-spanning primers and are listed in Appendix A. Rice ubiquitin 5 (Os01g22490) and elongation factor 1a (Os03g08050) were used as reference genes for qRT-PCR for the spatial-and developmental expression patterns and the hormone treatment-responsive expression patterns, respectively.

### 4.3. Gene Cloning

All total RNAs purified in this study were pooled to be used for the double-strand cDNA pools constructed using the SMART^™^ cDNA Library Construction Kit (Clontech, Mountain view, CA, USA) according to the manufacturer’s instruction. The predicted open reading frame (ORF) sequences were obtained from RAP-DB (https://rapdb.dna.affrc.go.jp/) and amplified from the ds-cDNA pools by Phusion^®^ High-Fidelity DNA polymerase (Toyobo, Osaka, Japan) with the primers containing an *att*B1 or *att*B2 sequence for creating the Gateway Entry clones (Appendix A). To identify their ORF sequences, the amplified PCR products were cloned into pDONR221 using Gateway^®^ BP Clonase^®^ II Enzyme Mix according to the manufacturer’s instructions. Then, those entry clones were further amplified by KOD FX polymerase (Toyobo, Osaka, Japan) with corresponding primers containing the recognition site of a *Nde*I restriction enzyme (Appendix A) to allow them to be cloned into pDONR221-Kz-*Nde*I-sGFP [38] for the analyses of their subcellular localizations in rice protoplasts. The amplified PCR products were digested with *Nde*I restriction enzyme (NEB, Ipswich, MA, USA) and cloned into pDONR221-Kz-*Nde*I-sGFP.

### 4.4. Subcellular Localization Analysis Using the Rice Greening Protoplasts

The preparation of rice protoplasts and their transfection using polyethylene glycol (PEG) were performed as previously described [38] using the leaves of 7 day-old-seedlings. The leaves of rice (*Oryza sativa* L. *Japonica* cv. “Ilmi”) seedlings were segmented to be digested in Cellulase R-10 and Macerozyme R-10 enzyme solution (Yakult Honsha, Japan) for the removal of cell walls. 10^6^ isolated protoplast cells were transfected with 5 μg of each plasmid with an equal volume of 40% PEG-3350 solution (Sigma, St. Louis, MO, USA) containing 0.5 M mannitol and 100 mM CaCl_2_. After the overnight incubation in the dark at 28 °C, fluorescence signals from the transfected protoplasts were observed and imaged using a Carl Zeiss LSM700 inverted confocal microscope (Carl Zeiss, Oberkochen, Germany).

The sub-organelle marker constructs OsPSY2-mCherry [38], G-rb and px-rb [39] were used to identify plastoglobules, golgi complexes and peroxisomes, respectively. Mitochondria were probed by MitoTracker^™^ Red CMXRos (Invitrogen, Carlsbad, CA, USA) and the auto-fluorescent signals of chlorophylls were detected at excitation and emission wavelengths of 555 and >650 nm, respectively.

## Figures and Tables

**Figure 1 ijms-21-08927-f001:**
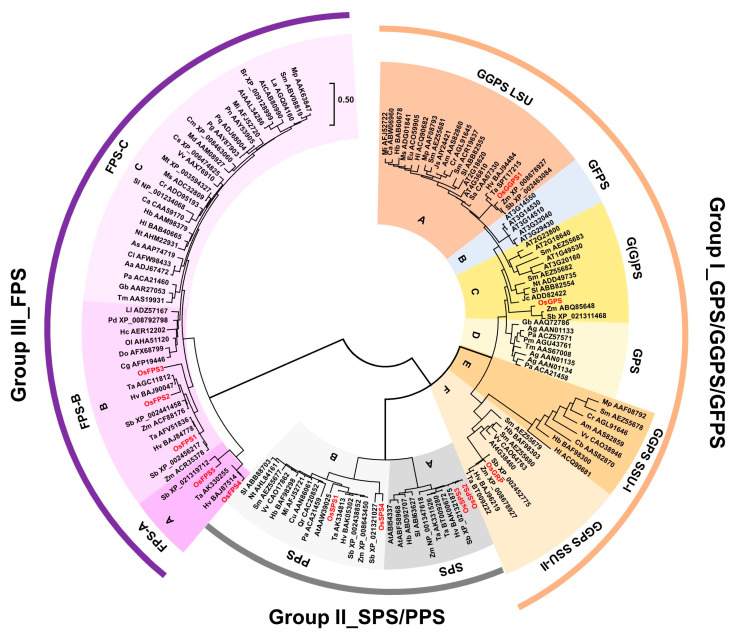
Evolutionary relationships among plant *trans*-PTs family. The deduced amino acid sequences of 136 plant *trans*-PTs genes were aligned using BioEdit 7.2.5 software with ClustalW algorithm and the phylogenetic tree was generated using MEGA 7.0.26 software by Maximum Likelihood tree method based on the ClustalW alignment. The scale bar represents the branch length at 0.50 substitutions per site. Plant PTs are named by the combination of a species abbreviation and an accession number and rice PTs are marked in red letters. All information of plant *trans*-PTs used in phylogenetic analysis is listed in Appendix A.

**Figure 2 ijms-21-08927-f002:**
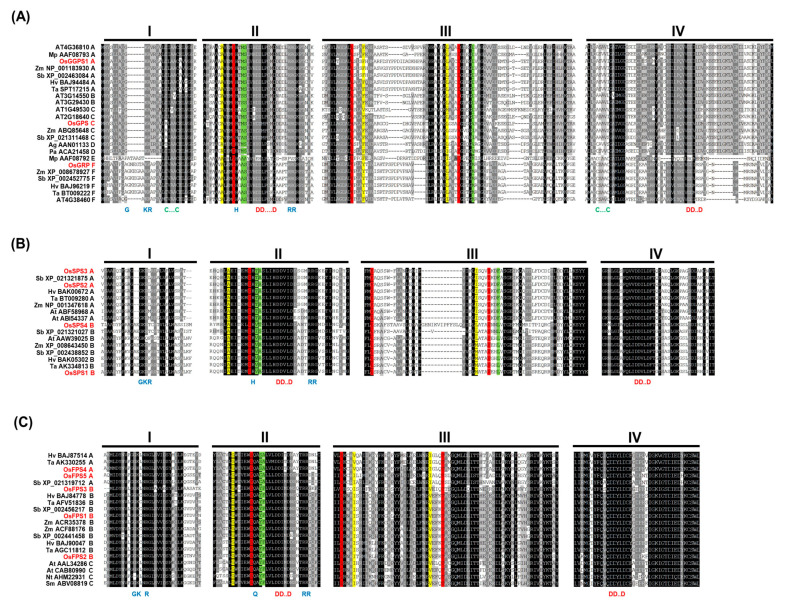
Multiple sequence alignment for plant *trans*-PTs. The deduced amino acids of the representative PTs in subgroups I–III were aligned using the standard parameters of the ClustalW algorithm, as shown in (**A**–**C**), respectively, and four kinds of highly conserved regions (**I**–**IV**) were displayed using GeneDoc program. The conserved amino acids are presented below with the blue letter, and the conserved sequences of FARM (first aspartate-rich motif) and SARM (second aspartate-rich motif) are marked with red letters. The floor 1, 2 and 3 residues of the “three floors” model [13] are colored green, red and yellow, respectively. The information of plant *trans*-PTs are listed in Appendix A.

**Figure 3 ijms-21-08927-f003:**
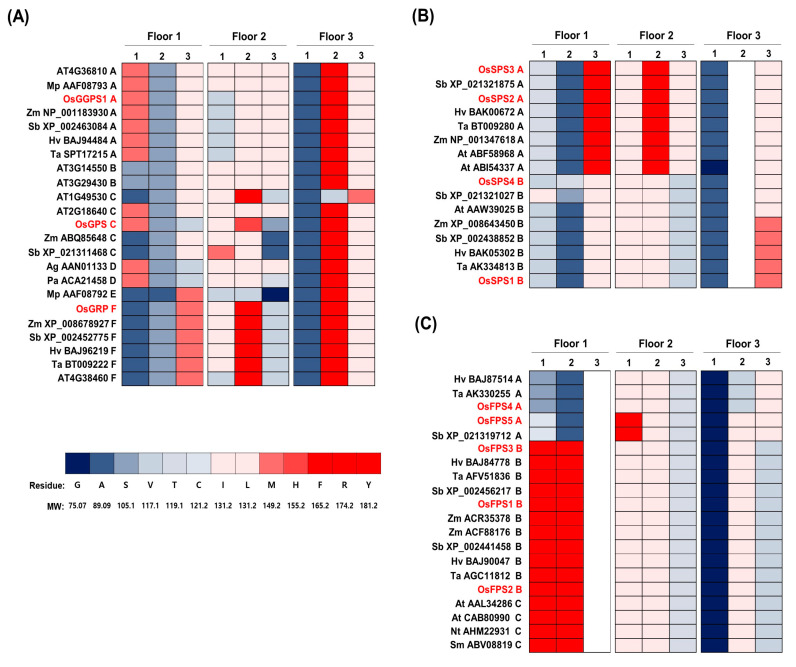
The “three floors” residues of plant *trans*-PTs. The amino acid residues of “three floors” are presented as different ranged-colors by following their molecular weights in the HeatMap graphs of Group I (**A**), Group II (**B**) and Group III (**C**). A white blank means missing amino acid residues. The “three floors” residues were listed with the properties of amino acids Appendix A.

**Figure 4 ijms-21-08927-f004:**
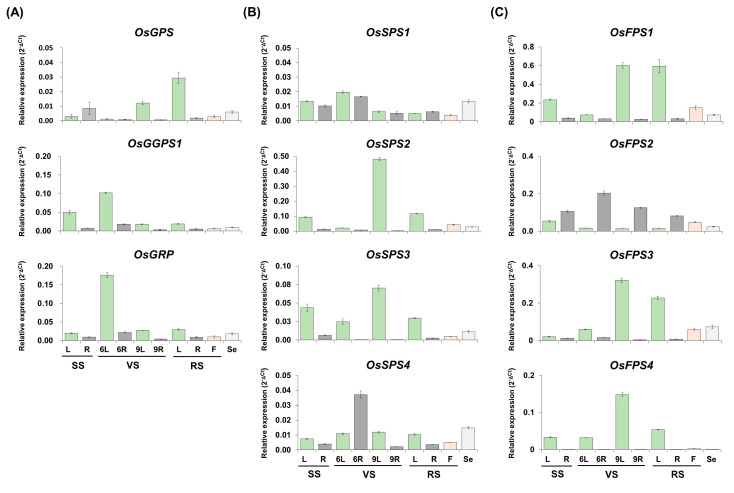
Spatial and temporal expression of rice *trans*-PT genes. The endogenous transcript levels of rice *trans*-PT genes in Group I (**A**), Group II (**B**) and Group III (**C**) were examined by qRT-PCR using total RNA isolated from various tissues, including leaves (L) and roots (R) of seedling stage (SS), vegetative stage (VS) and reproductive stage (RS) tissues and florets (F) of RS and seeds (Se) harvested at 40 days after flowering, using gene-specific primers as shown in Appendix A. All transcript levels were measured in three technical replicates and were calculated by the ΔCt equation against the *OsUbi5* gene. The results are expressed as the mean ± standard error (SE) and the values are listed in Appendix A.

**Figure 5 ijms-21-08927-f005:**
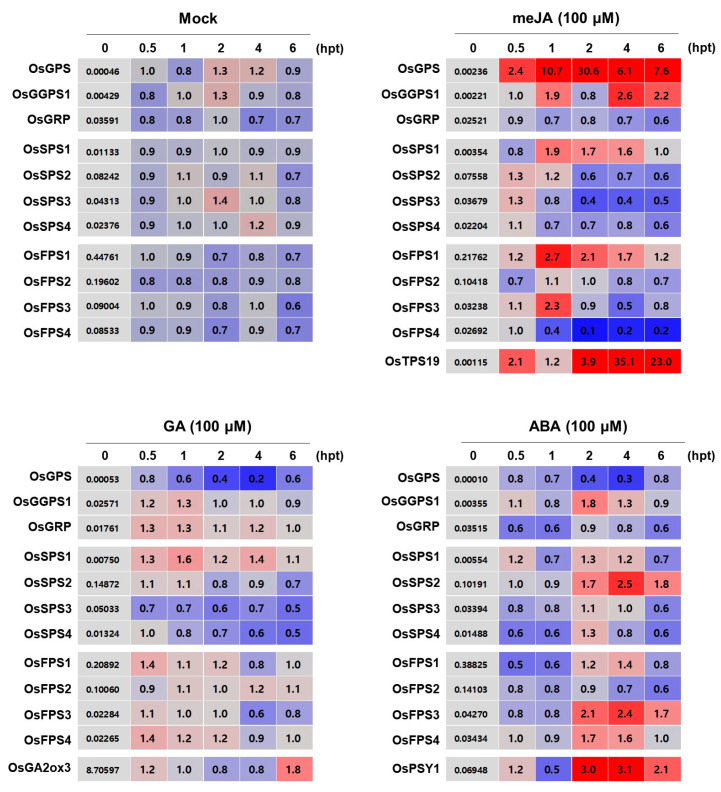
Expression profiles of rice *trans*-PTs in response to different hormone treatments in 10-day-old leaves. The heatmap graphs are presented by the relative fold-change expression in comparison to their respective controls. The ΔCt values of each time point (0, 0.5, 1, 2, 4 or 6 h) post treatment (hpt) were calculated by the ΔCt equation against the *OsEF1**α* gene and the relative fold-change values (2^−^^ΔΔCt^) of each time point were obtained by the normalization using the ΔCt values of each 0 h time point as a respective control. The 0 h time points of each treatment are presented using the ΔCt values in the gray colored boxes and other time points are displayed using the fold change values. The color key of the heatmap graph is presented under the last graph. The results are expressed as the mean values of three technical repeats and also listed in Appendix A. Mock, with no chemical; meJA, methyl jasmonic acid; GA, gibberellic acid; ABA, abscisic acid.

**Figure 6 ijms-21-08927-f006:**
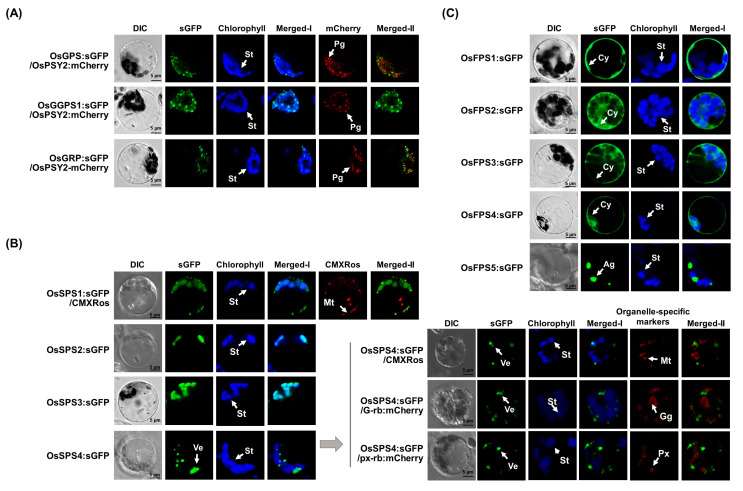
Subcellular localization of rice *trans*-PTs. Green fluorescent protein (sGFP) was fused to rice *trans*-PTs of Group I (**A**), Group II (**B**) and Group III (**C**) and sGFP-fused constructs were transfected into rice protoplasts of 10-day-old leaves. The images were acquired at 630× magnification using a confocal microscope. The white signals of Merged-I are derived from overlap of the green sGFP-fluorescence signal and the blue chlorophyll autofluorescence. The mCherry fused construct of *O. sativa* phytoene synthase 2 (OsPSY2) was used as a control for plastoglobules (Pgs) localization and MitoTracker™ Red CMXRos as a red-fluorescent dye was used to stain mitochondria in rice protoplasts, the mCherry fused constructs of G-rb and px-rb were used as a control for Golgi apparatus and peroxisome, respectively [39]. Individual merged-II showed the overlap of the green from sGFP and the red fluorescent signals derived from CMXRos or mCherry. Ag, aggregates; Cy, cytosol; Gg, Golgi; Mt, mitochondria; Pg, plastoglobules;Px, peroxisomes; St, stroma; Ve, vesicles.

**Figure 7 ijms-21-08927-f007:**
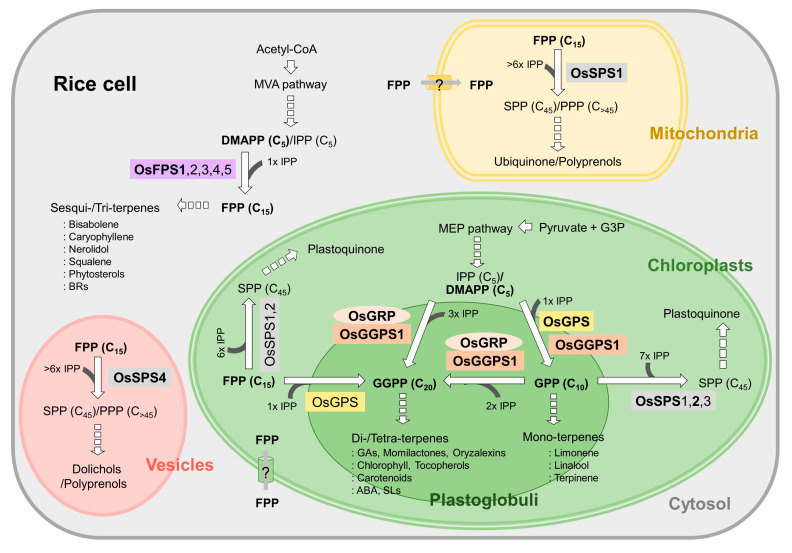
The networking model of rice *trans*-PTs based on subcellular topology. The subcellular organelles are colored: Chloroplasts with plastoglobules (green with dark green), mitochondria (beige) and cytosol with vesicles (light-gray with pink). Rice *trans*-PTs have colored background in yellow (OsGPS), orange (OsGGPS1), pale orange (OsGRP), grey (OsSPSs) and purple (OsFPSs). DMAPP, dimethylallyl pyrophosphate; FPP, farnesyl pyrophosphate; FPS, FPP synthase; GFPP, geranylfarnesyl pyrophosphate; GFPS, GFPP synthase; GPP, geranyl pyrophosphate; GPS, GPP synthase; GGPP, geranylgeranyl pyrophosphate; GGPS, GGPP synthase; PPP, polyprenyl pyrophosphate; PPS, PPP synthase; SPP, solanesyl pyrophosphate; SPS, SPP synthase. Allelic substrates and main actors among PTs with the same enzymatic activity are bolded.

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
