# Peer review of "The Predicted Functional Compartmentation of Rice Terpenoid Metabolism by Trans-Prenyltransferase Structural Analysis, Expression and Localization"

_ijms, 2020, doi:10.3390/ijms21238927_

Round 1

Reviewer 1 Report

The authors have made great efforts to elucidate the biochemical functions of the trans-PT genes from the rice whole-genome database, construct the phylogenetic tree and predict biochemical functions by matching sequences. The attained results are of interest to the readers. The paper has been well documented and written as a scientific paper. Some minor comments and suggestions may be useful for authors to improve the quality of the paper:

  1. Some information should be added the references to confirm (see in the pdf file)
  2. In the result part, some sentences seem to be not results which should be moved to the discussion part; for example page 3 line 95; page 7 line 190-191, etc (see the pdf file)
  3. Some minor spelling and typos should be checked (see in the file); for example page 5 line 151; page 9 line 242 etc
  4. Figure 4, page 9, the standard error bar should be added the specific value
  5. The conclusion should be reworded to highlight the significant results of this work;
  6. In Materials and Methods; page 17 line 411, the specific name of Korean rice should be defined;
  7. The references should be rechecked, such as page number and abbreviation of journal names

Author Response

We would like to take this opportunity to express our thanks to the reviewers for the positive feedback and helpful comments for correction or modification, and we believe to have resulted in an improved revised manuscript, which you will find uploaded alongside this document.

The manuscript has been revised to address the reviewer comments, which are appended alongside our responses to this letter.

We sincerely appreciate that our manuscript has been greatly improved by the excellent comments of our reviewers, and we believe that the revised manuscript is suitable for publication in “International Journal of Molecular Sciences”.

Sincerely yours,

Reviewer 1

  1. Some information should be added to the references to confirm.
    : It has been revised, as follows
    In page 2 line 62-65:
    “Furthermore, the ‘three floors’ model has been proposed as a very useful tool to elucidate the trans-isoprenyl-PP elongation mechanisms and to predict the characteristic activity of trans-PTs in determining the chain length of the linear isoprenoid backbone [13]”

  2. In the result part, some sentences seem to be not results which should be moved to the discussion part; for example, page 3 line 95; page 3 line 95, etc (see the pdf file).
    : It has been revised, as follows
    In page 3 line 94-95 for page 3 line 190:
    “Twelve genes of rice trans-PTs were identified from the rice whole genome database, through the BLAST (Basic Local Alignment Search Tool) search engine.”

    è In page 3 line 181-182 for page 3 line 190:
    “Lastly, two CxxxC motifs were presented only in Group I, but not in Groups II and III (Figure 2).”
    : the explain of CxxxC motif with a reference was moved to introduction part in page 2 line 61-63
    è In page 2 line 60-61 for page 3 line 190:
    “CxxxC (x is any hydrophobic amino acid residue) motifs critical for the physical interaction between GGPS_LSU and GGPS_SSU families to alter or enhance GPS/GGPS enzyme activities [10, 11].”

    è In page 9 line 244-247 for page 10 line 254:
    “Since the production of terpene metabolites have been reported to be closely related to responses against various plant hormones, such as methyl jasmonate (meJA) [30, 31], gibberellic acid (GA) [32] and abscisic acid (ABA) [33], the transcriptional response of rice trans-PTs to meJA, GA and ABA treatments was investigated.”

  3. Some minor spelling and typos should be checked (see in the file); for example, page 5 line 151; page 9 line 242 etc
    : It has been revised, as follows
    è In page 5 line143 for page 5 line 151:
    “the independent evolution in the plant kingdom is indicated”

    In page 8 line 230-232 for page 9 line 242:
    “OsSPS3 in subgroup A were expressed mainly in leaves of all developmental stages, with the highest peaks at the late vegetative stage (Figure 4B), showing similar expression patterns within the same subgroup.”

  4. Figure 4, page 9, the standard error bar should be added the specific value
    : It has been revised, as follows
    The specific values of standard errors are listed in Table S6, which has been newly updated in the revised manuscript.

  5. The conclusion should be reworded to highlight the significant results of this work
    : It has been added, as follows
    Added in page 14 line 407-411:
    “In summaries, all of the rice trans-PT genes through construction of a phylogenetic tree using 136 plant trans-PT genes were collected, and their biochemical functions were predicted by matching sequences with known conserved motifs and the ‘three floors’ model, taken together with their expression patterns and subcellular localizations to suggest a topology-based working model of trans-PT for understanding the predicted-functional compartmentation of rice terpenoid metabolism.”

  6. In Materials and Methods; page 17 line 411, the specific name of Korean rice should be defined
    : It has been revised, as follows
    In page 15 line 425 for page 17 line 411:
    “Mature seeds of Korean rice ‘Ilmi’ (Oryza sativa) were sterilized with…”

  7. The references should be rechecked, such as page number and abbreviation of journal names
    : It has been completely revised.

Reviewer 2 Report

In the MS “The Predicted-Functional Compartmentation of Rice Terpenoid Metabolism Predicted by trans-Prenyltransferase Structural Analysis, Expression, and Localization”, the authors analyze the sequences, expression pattern, and the localization of trans-Prenyltransferase genes. There is limited information about the function of these genes. It is better to do some enzyme activity analysis and functional analysis. The current version of this paper does not meet the standard published on IJMS.

How to determine the concentration of MeJA, GA and ABA? there are all 100um?

The length to width ratio of the localization pictures seems to be abnormal and it seems to be compressed. Please add error bar for pictures.

Gene and Arabidopsis should be Italic.

Author Response

Reviewer 2

  1. The authors analyze the sequences, expression pattern, and the localization of trans-Prenyltransferase genes. There is limited information about the function of these genes. It is better to do some enzyme activity analysis and functional analysis.
    è Please, consider the following points.
    : There has been no comprehensive reports of all rice trans-PTases, although the biochemical functions of OsGPS, OsGGPS1, OsGRP, OsSPS1 and OsSPS2 were previously reported. So, our research goal was to comprehensively understand the possible roles of rice trans-PTases rice terpenoid metabolism. For this, we have tried to collect possible information of all rice trans-PTases together with our expression profiling data and subcellular localization data, and comprehensively considered their working model on rice terpenoid metabolism to predict the topology-based working model of them in this manuscript.
    è We agree with your comments that the biochemical functions of rice FPSs should be newly elucidated, since it has not been studied yet. However, to complete the biochemical functions of rice FPSs, the intensive biochemical analyses under in vitro and in vivo conditions should be performed. So, we have thought that we should find another way in this manuscript, even if the deduced results will be good for our next studies.
    è Similarly, other reviewers also commented that the description of rice FPS about the results of Figure 3 was overinterpreted. We sincerely agree with their opinions, and have revised the interpretation of Group III_A to be constricted: In page 7 line 216-219:
    “In Group III, the first two residues of Floor 1 in subgroup A, which includes OsFPS4 and 5, were observed to be quite different as the light residues S and A (MW: <105.1), compared with the heavy residues of F and Y in subgroup B and C (Figure 3C).”
    è If the overinterpretation of Group III could be reduced, we are carefully thinking that the topology-based working model of rice PTases could be accomplished without any enzyme assays of OsFPSs.
    è Please, consider the aboves.

  2. How to determine the concentration of MeJA, GA and ABA? there are all 100um?
    : The description of experimental procedure has been further updated to easily figure-out the entire experimental procedures in detail, as follows
    è In page 16 line 432-443:
    “The sterilized seeds were grown on MS agar medium for 3 days, and healthy seedling plants were transplanted and grown on four layers of gauze (5 × 5 cm) in the Incu tissue culture frames (model 310071; SPL Lifesciences, Seoul, Korea) containing 45 ml of liquid MS medium (Duchefa, Haarlem, Netherlands) under 16 h light/8 h dark at 28°C for 7 days, which were prepared by filling the culture frames containing four layers of gauze (5 × 5 cm) with the liquid MS medium (Duchefa) up to the 45 ml-marked line and sterilizing them by autoclave. Plant hormones methyl jasmonic acid (MeJA), gibberellic acid (GA) and abscisic acid (ABA) were prepared as 10 × stock solution of 1 mM, and after checking whether 45 ml-liquid MS media were filled in the culture frames, 5 ml of 1 mM hormone solution (10 ×) were added to each 45 ml of liquid MS medium with gentle stirring. For the 10-day old seedling plants that had been grown on gauze, whole plants containing leaves and roots were sampled at 0, 30 min, 1, 2, 4 and 6 hours post treatment into liquid nitrogen for the expression pattern analysis of rice PTs genes.”

  3. The length to width ratio of the localization pictures seems to be abnormal and it seems to be compressed.
    : It has been fixed in the revised manuscript.

  4. Please add error bar for pictures
    : It has been revised, as follows
    è The specific values of standard errors are listed in Table S6, which has been newly updated in the revised manuscript.

  5. Gene and Arabidopsis should be Italic.
    : It has been completely revised.

Reviewer 3 Report

Experiments were well designed. Results and data are clear and believable.

Few comments:

There are no correlation data between expression patterns (Figure 5) and localization (Figure 6) of trans-PTs in current version. I am wondering what happens in the localization patterns of trans-PTs of Group I (A), Group II (B) and Group III after MeJA, GA and, ABA treatment? Please add these data in revised version if authors have any result or mention expected data in Discussion section.

OsSPS4 and OsFPS2 are mainly expressed in the root (Figure 4B and C). But, authors examined their expression using only leaf samples after MeJA, GA and, ABA treatment (Figure 5). Please describe the reason or need to discuss about this.

Author Response

  1. Reviewer 3

    1. There are no correlation data between expression patterns (Figure 5) and localization (Figure 6) of trans-PTs in current version. I am wondering what happens in the localization patterns of trans-PTs of Group I (A), Group II (B), and Group III after MeJA, GA and, ABA treatment? Please, add these data in revised version if authors have any result or mention expected data in Discussion section
      : The description about the roles of rice PTases in supplying the primary substrates to terpene biosynthetic enzymes has been added to the revised manuscript, as follows
      è In page 13 to 14 line 361-393,
      “Taken together, we have further considered the important roles of rice PTases in supplying the primary substrates (GPP/FPP/GGPP) to terpene biosynthetic enzymes. As shown in Figure 5, the expression of OsTPS19 is strongly increased by meJA-treatment with OsGPS………. The critical roles of rice PTases in terpenoid metabolism should be further considered with the biosynthetic sites of the above terpene metabolites, in the near future.”

    2. OsSPS4 and OsFPS2 are mainly expressed in the root (Figure 4B and C). But, authors examined their expression using only leaf samples after MeJA, GA and, ABA treatment (Figure 5). Please describe the reason or need to discuss about this
      : It has been revised, as follows
      è In page 16 line 441-442:
      “whole plants containing leaves and roots were sampled ……..”

Reviewer 4 Report

The manuscript describes phylogenetic, bioinformatic, and biological analysis of rice trans-prenyltransferases, and proposes the topology-based working model. Comprehensive analysis of rice trans-prenyltransferase will have potential for wider investigations of rice terpene metabolites.  However, I do not know why the phylogenetic relationship was reconstructed by UPGMA method. Recently, most of the phylogenetic analysis have been performed by most likelihood and/or Bayesian tree. Moreover, authors seem to stretch the interpretation of a ‘three floor’ model. For example, authors suggested distinct roles between rice and other plant FPS, resulting from sequence comparison in the model. Because Wang et al. proposed this model to explain the product chain-length determination of short chain trans-prenyltransferases (Ref. 23), this model could not come to the conclusion that distinct roles between paralogous prenyltransferases result from different amino acid residues in each ‘floor’. Finally, to reinforce the conclusion, biochemical characterization of newly identified prenyltransferase genes should be carried out.

Author Response

  1. Reviewer 4

    1. The reason why the phylogenetic relationship was reconstructed by UPGMA method. Recently, most of the phylogenetic analysis have been performed by most likelihood and/or Bayesian tree.
      è During the revision, we performed Maximum likelihood tree analysis of rice PTases using MEGA 7.0.26 software, according to the reviewer comments.
      è As the results, the phylogenetic tree showed similar patterns in grouping the major groups (Group-I/II/III) and subgroups (seven in Group I, two in Group II, and three in Group III), except that the positions of two subgroups were reversed in order.
      è Although the UPGMA method was out of date, we are carefully thinking that it still be suitable for our study, since the results are in our research purpose category. So, we would like to keep the results of Figure 1,2,3 continuously in the revised manuscript.
      è Please, consider the above reasons.

    2. The over-interpretation of a ‘three floor’ model about proposing distinct roles between rice and other plant FPS
      : It has been revised, as follows
      è In page 7 line 216-219:
      “In Group III, the first two residues of Floor 1 in subgroup A, which includes OsFPS4 and 5, were observed to be quite different as the light residues S and A (MW: <105.1), compared with the heavy residues of F and Y in subgroup B and C (Figure 3C).”
      : the interpretation of group III_A was revised to be constricted to convey the important meaning of the results in Figure 3 more clearly.

Round 2

Reviewer 2 Report

It is better to do some enzyme activity analysis and functional analysis. 

Author Response

Reviewer 2

  1. It is better to do some enzyme activity analysis and functional analysis.

    : To satisfy your comment and editor’s comment “ The authors need to describe some explanations in the manuscript (in results or discussion) on enzyme activity”, we add the information about the identification of rice PTs enzyme activity in Table 1, and the corresponding descriptions were added as follows;

    è In page 13 line 349 to 355
    : “As summarized in Table 1, GPP/GGPP can be produced by both OsGPS and OsGGPS1 in rice. OsGPS not only biosynthesize mainly GPP, but can also produce GGPP under the limited conditions in which FPP is present as a substrate, and likewise, OsGGPS1 can also produce the limited amount of GPP in addition to GGPP [10]. OsSPS1 and OsSPS2 were reported to produce SPP/PPP by using the primary substrates such as GPP, FPP and GGPP [20], OsSPS3 [21] and OsSPS4 were identified as the putative SPS family proteins, and the biosynthesis of FPP was predicted to be mediated by five kinds of OsFPSs in this study and two previous studies [22, 24].”

    : To more tone down the prediction of rice PTs enzyme activities, several descriptions in discussion section were modified as follows;

    è In page 11-12 line 295 to 312
    : “In our phylogenetic tree, GGPS/GGPS/GFPS family (Group I) was subdivided into six subgroups (Figure 1). Rice has relatively simpler system in Group I, with only three PTs, compared to Arabidopsis with twelve PTs. Previously in rice, OsGGPS1 (LSU) was reported to have multiple functions as GGPS (67%), GPS (27.2%) and FPS (5.5%) as a homomeric complex, and OsGRP (SSU-II) strongly enhanced GGPS function as heteromeric complex with OsGGPS1 [10]. In previous reports, the heteromeric LSU-SSU complexes mediated the enhancement of GPP-biosynthetic activity to produce both GPP and GGPP using DMAPP, GPP, and FPP in Arabidopsis [15] and hops (Humulus lupulus; Wang and Dixon [11]), or enabled GGPS_LSU to biosynthesize only GPP without any GGPP-production, even if the homomeric LSU complexes had GGPP-biosynthetic function, as in snapdragon (Antirrhinum majus, Tholl et al. [40]), or no enzymatic function, as in peppermint (Mentha x piperita, Chang et al. [23]), and in those plant system, any homomeric GPS has been not reported yet.”

    è In page 14-15 line 382 to 402
    : “In summaries, all rice trans-PT genes through a phylogenetic tree using 136 plant trans-PT genes were collected, and their biochemical functions were predicted by matching sequences with known conserved motifs and the ‘three floors’ model, taken together with their expression patterns and subcellular localizations. Collectively, we propose a predicted topology-based working model of rice trans-PTs to biosynthesize the linear trans-isoprenyl-PP backbones in Figure 7, by the following points: where rice PTs are distributed in rice cells, what kinds of allylic substrates they utilize, and what kinds of terpene metabolites are produced in rice. As the major cellular-compartmentations of trans-isoprenyl PPs, GPPs for mono-terpenoids are produced by OsGPS in plastoglobules or OsGGPS1 in stroma, GGPPs for di-/tetra-terpenoids are produced by a heteromeric complex between OsGGPS1 and OsGRP or a homomeric OsGGPS1 in both plastoglobules and stroma, SPP/PPP are by OsSPS2 in stroma for plastoquinone, by OsSPS1 in mitochondria for ubiquinone/polyprenols, and by OsSPS4 in cytosolic vesicles for dolichols/polyprenols, and FPPs are by OsFPS1 in cytoplasm for sesqui-/tri-terpenoids (Figure 7). In rice, the diverse terpene metabolites have been reported, which include phytohormones, chlorophyll, tocopherols, carotenoids, plastoquinone, ubiquinone, phytosterols, and dolichols to play diverse essential roles for plant development, and for adaptation to environmental conditions, antimicrobial monoterpenes such as limonene, linalool and terpinene, and other anti-fungal and insect- attractant terpenes such as momilactones, phytoallexins (oryzalexins), bisabolene, caryophyllene,and nerolidol (bactericides) [45-48]. Our networking model of all predictable rice PTs based on subcellular topology expands our understanding about rice terpenoid metabolism, and if combined with the further identification of enzymatic activity in the near future, it could be very useful to apply to functional terpenoid metabolic engineering in monocot plant systems, including rice.”
